# Enhancing Mathematical Reasoning in Language Models Through Focused Differentiation Training

## Abstract

Enhancing the mathematical capabilities of large language models (LLMs) is crucial for applications requiring precise and rigorous mathematical reasoning. Current models, even when trained with methods like Direct Preference Optimization (DPO), often struggle to effectively differentiate between correct and erroneous mathematical responses, especially when errors occur in multi-step solutions. Traditional approaches focusing on token or logit-level analysis fail to capture the nuanced semantic differences in mathematical reasoning. To address this challenge, we propose leveraging the rich semantic information embedded in the hidden state space of LLMs. Our novel approach, Focused Differentiation Training (FDT), fine-tunes the model by emphasizing the differences between the hidden states of correct and incorrect responses, rather than their common features. Unlike other methods that detect errors at the token or logits level and often rely on human input or more powerful models, our approach enhances mathematical reasoning capabilities using only the model's inherent abilities. This methodology promotes a more accurate alignment with mathematical correctness, thereby improving the model's ability to evaluate and generate precise mathematical responses. Experimental results demonstrate that our algorithm substantially outperforms traditional alignment methods in mathematical tasks, offering a robust solution for enhancing the mathematical reasoning capabilities of language models.

## 1 Introduction

Reinforcement Learning from Human Feedback (RLHF) has emerged as a powerful technique for aligning large language models (LLMs) with human preferences, significantly enhancing their usability and reliability across various applications. While traditional approaches to training LLMs often rely on vast amounts of data and heuristic methods, potentially leading to misalignment with human values and intentions, RLHF addresses this issue by directly incorporating human feedback into the training process, thereby ensuring that models better reflect human preferences.

Existing methods for RLHF, such as Direct Preference Optimization (DPO), have demonstrated significant improvements in aligning LLM outputs with human preferences in general language tasks. However, when it comes to mathematical reasoning tasks, especially those involving multi-step problems, these methods often fall short. Traditional RLHF approaches typically focus on general language understanding and response generation, which may not adequately capture the specific nuances required for precise mathematical task performance. This oversight can result in models that, although aligned with general linguistic preferences, often fail to address the structured and logical demands unique to mathematical reasoning and problem-solving effectively.

The primary challenge with existing RLHF techniques in mathematical contexts is their inability to accurately identify and differentiate errors within multi-step solutions. For instance, if a response contains a single incorrect step amidst otherwise correct reasoning, it is often labeled entirely wrong. This labeling approach discourages models from recognizing the complexity and partial correctness within mathematical arguments, a critical skill for advanced mathematical reasoning. Furthermore, current methods rely heavily on token or logit-level analysis, which often proves inadequate for capturing the subtle semantic differences crucial in mathematical reasoning **?**. These methods rely

on the assistance from human or more advanced models, which is costly and labor-intensive. Additionally, these approaches may not always provide consistent or reliable interpretations, especially in complex multi-step mathematical problems where nuanced understanding is essential.

To address these shortcomings, we propose leveraging the hidden states of LLMs to more effectively differentiate between correct and incorrect mathematical responses. The hidden state space of LLMs contains rich semantic information that is more concentrated and intact compared to surface-level token representations. By focusing on this dense embedding space, we can capture and utilize semantic divergences more effectively, allowing for a deeper analysis of mathematical reasoning processes. Our approach is inspired by recent advancements in semantic analysis of embedding spaces, as described by **?**. This method ensures that the model engages with underlying semantic structures rather than merely responding to explicit linguistic cues. Moreover, as highlighted by **?**, representing semantic divergence in embedding space provides a promising solution to the problem of semantic equivalence and linguistic invariances, which are particularly relevant in mathematical contexts.

Building on these insights, we introduce a novel training algorithm called Focused Differentiation Training (FDT). FDT operates by fine-tuning the weight updates in the model's output layer, with a specific emphasis on distinguishing the differences between correct and incorrect mathematical responses rather than their common features. This approach is grounded in the observation that the common parts of hidden states between correct and incorrect answers often represent shared mathematical concepts or problem setups, while the differences are more likely to indicate critical points of divergence in reasoning. By specifically targeting how the model perceives and processes mathematical logic through hidden state analysis, FDT aims to enhance the model's ability to dissect and understand the underlying mathematical structure. This method can be seamlessly integrated into existing RLHF frameworks, thereby improving the model's performance on tasks that require high levels of mathematical accuracy and reasoning. The key contributions of our work are as follows:

- We introduce FDT, a novel algorithm that leverages hidden state analysis to fine-tune the weight updates in the model's output layer, enhancing mathematical reasoning capabilities. This method can be plugged into existing RLHF frameworks to improve the model's ability to distinguish between correct and incorrect mathematical responses, particularly in multi-step problems.

- We present a theoretical analysis of how FDT improves the model's ability to distinguish between correct and incorrect mathematical responses.

- We provide empirical evidence demonstrating the superiority of FDT over traditional RLHF methods in mathematical reasoning tasks, showcasing significant improvements in accuracy over a range of mathematical tasks and several models.

## 2 RELATED WORK

### 2.1 REINFORCEMENT LEARNING FROM HUMAN FEEDBACK

Reinforcement Learning from Human Feedback (RLHF) has become a central approach for aligning large language models (LLMs) with human values by incorporating human evaluations to refine model outputs iteratively (**?????**). Unlike traditional reinforcement learning, which relies on predefined rewards, RLHF uses qualitative feedback from human evaluators to guide the model toward more human-like and ethical responses. However, its implementation poses challenges due to the variability and subjectivity of human-generated feedback, which can introduce inconsistencies into the reward model (**?**).

Due to the various limitations of RLHF, researchers have started exploring new paradigms for aligning large models. In particular, DPO (**?**) marks a significant advancement in direct policy optimization, addressing the complexities of balancing model behavior through a more refined approach to reward function optimization. Subsequently, numerous variants of DPO have emerged. SimPO (**?**) observed that during DPO training, the curve representing the change in probabilities for the model's generated responses does not align with the implicit reward curve. To address this, SimPO proposed directly amplifying the probability gap between the chosen and rejected responses. KTO (**?**) re-

structured DPO's loss function, removing the dependence on pairwise datasets during the alignment process. TDPO (**?**) re-derived the RLHF problem from a token-level perspective, achieving a better balance between model alignment and generation diversity. ORPO (**?**), from a more lightweight perspective, further eliminated the reliance on reference models during the alignment process.

## 2.2 MATHEMATICAL REASONING

Large language models (LLMs) have exhibited substantial mathematical reasoning abilities. However, when faced with complex mathematical problems that require fine-grained reasoning, LLMs still struggle to perform effectively. In such cases, LLMs may even exhibit severe hallucination issues. One common approach to addressing this issue is to impose stricter constraints on the model by requiring more detailed, step-by-step reasoning, thereby enhancing the model's Chain-of-Thought (CoT) capabilities (**?????**). While this approach has proven effective in certain tasks, it does not fundamentally improve the model. Moreover, due to the inherent limitations of the model's architecture, the potential improvements are quite limited. When presented with questions in different formats, the model's responses can still display hallucinations, indicating that the root cause of the hallucination problem has not been addressed.

Another approach focuses on significantly improving the model's mathematical reasoning capabilities through continued pre-training (CPT) or supervised fine-tuning (SFT) on large-scale, high-quality mathematics-related datasets (**????????**). During this process, various data augmentation techniques, such as rephrasing, expansion, and evolution, are widely applied to further enrich the datasets, helping the model achieve better performance during CPT or SFT. While, these datasets are collected off-policy with respect to the model itself, which limits their ability to correct some of the model's intrinsic errors.

Reinforcement learning (RL) is another class of methods that can significantly enhance the logical reasoning capabilities of LLMs (**???**). By progressively strengthening the model's reasoning abilities and reducing hallucinations during inference, RL improves the reliability of the reasoning process. Recent studies have shown that combining reinforcement learning with mathematical reasoning tasks can effectively improve the model's accuracy, particularly for complex mathematical problems. This category of methods includes RLHF, DPO, and DPO-like approaches. Among these, Step-DPO (**?**), a DPO-like method, stands out for its ability to significantly enhance mathematical reasoning by aligning the reasoning process step-by-step in long-chain reasoning tasks, thereby correcting specific errors in LLMs' mathematical reasoning.

## 3 PRELIMINARIES

In language generation tasks, a language model (LM) is provided with a prompt (denoted as $x$) to produce a corresponding response (denoted as $y$), where both $x$ and $y$ are represented as token sequences. Direct Preference Optimization (DPO) builds on the reinforcement learning (RL) objective used in Reinforcement Learning with Human Feedback (RLHF):

$$\max_{\pi_\theta} \mathbb{E}_{x \sim \mathcal{D}, y \sim \pi_\theta(\cdot|x)} \big[ r(x, y) - \beta D_{\mathrm{KL}} \big( \pi_\theta(\cdot \mid x) \big\| \pi_{\mathrm{ref}}(\cdot \mid x) \big) \big], \tag{1}$$

where $\mathcal{D}$ stands for the human preference dataset, $r(x, y)$ represents the reward function. The reference model, denoted as $\pi_{\mathrm{ref}}(\cdot|x)$, typically selects the language model after supervised fine-tuning. $\pi_\theta$ refers to the model undergoing RL fine-tuning. $\beta$ corresponds to the coefficient applied to the reverse KL divergence penalty.

To better align the model's output with human preferences, DPO employs the Bradley-Terry model for conducting pairwise comparisons:

$$P_{\mathrm{BT}}(y_1 \succ y_2|x) = \frac{\exp(r(x, y_1))}{\exp(r(x, y_1)) + \exp(r(x, y_2))}. \tag{2}$$

By obtaining the closed-form solution for the reward model $r(x, y)$ and policy $\pi_\theta$ from Eq 1 and substituting it into the Bradley-Terry model, DPO derives the following loss function:

$$\mathcal{L}_{\text{DPO}}(\pi_\theta; \pi_{\text{ref}}) = -\mathbb{E}_{(x,y_w,y_l)\sim\mathcal{D}} \left[ \log \sigma \left( \beta \log \frac{\pi_\theta(y_w \mid x)}{\pi_{\text{ref}}(y_w \mid x)} - \beta \log \frac{\pi_\theta(y_l \mid x)}{\pi_{\text{ref}}(y_l \mid x)} \right) \right], \quad (3)$$

where $y_w$ and $y_l$ denote the preferred and dispreferred completion.

To maximize the logical reasoning capabilities of LLMs, Step-DPO (**?**) models the answers $y$ to long-chain mathematical problems as a sequence of reasoning steps $y = s_1, s_2, \ldots, s_n$, where $s_i$ is the $i$-th reasoning step. At each stage, given a prompt $x$ and the same correct reasoning steps $s_{1\sim k-1}$, Step-DPO aims to maximize the probability difference between the correct next reasoning step $s_{win}$ and the incorrect next reasoning step $s_{lose}$:

$$\mathcal{L}_{\text{Step}-\text{DPO}}(\pi_\theta; \pi_{\text{ref}})$$

$$= -\mathbb{E}_{(x,s_{1\sim k-1},s_{win},s_{lose})\sim\mathcal{D}} \left[ \log \sigma \left( \beta \log \frac{\pi_\theta(s_{win}|x,s_{1\sim k-1})}{\pi_{\text{ref}}(s_{win}|x,s_{1\sim k-1})} - \beta \log \frac{\pi_\theta(s_{lose}|x,s_{1\sim k-1})}{\pi_{\text{ref}}(s_{lose}|x,s_{1\sim k-1})} \right) \right].$$
$$(4)$$

Due to its unique structure, Step-DPO strictly relies on the dataset. To address this, Step-DPO utilizes the Chain-of-Thought (CoT) (**?**) method to collect an additional preference dataset[1]. The construction of the dataset relies on the human user or GPT-4 to identify the incorrect reasoning steps in the dataset, which are then used to train the model. However, this method is not always feasible, as it requires a large amount of human effort and may not always be reliable.

## 4 METHODOLOGY

In this section, we introduce the FDT algorithm, which fine-tunes the weight updates in the model's output layer to enhance mathematical reasoning capabilities. FDT aims to improve the model's ability to distinguish between correct and incorrect mathematical responses by focusing on the semantic divergence within the dense embedding space. We provide a detailed description of the FDT algorithm and its implementation in the context of mathematical reasoning tasks.

We first introduce and define the notation used throughout our theoretical analysis and the description of the DPO loss function. The input to the model is denoted by $x$. This prompt forms the basis from which both correct and incorrect responses are generated, represented as $y_w$ and $y_l$, respectively. We consider the response as a sequence of tokens $y^{1:T} = [y^1, y^2, ..., y^T]$, where $y^k$ represents the $k$-th token in the response $y^{1:T}$. If the length of the response $T'$ is shorter than $T$, we assume the $y^{T':T}$ is the padding token. Additionally, we assume that $y^0 = []$. The model's predicted probability of generating response $y$ given input $x$ is denoted by $\pi_\theta(y|x) = \prod_{t=1}^{T-1} \pi_\theta(y^{t+1}|x, y^{1:t})$, where $\theta$ represents the model's parameters. The reference model's predicted probability of generating response $y$ given input $x$ is denoted by $\pi_{\text{ref}}(y|x) = \prod_{t=1}^{T-1} \pi_{\text{ref}}(y^{t+1}|x, y^{1:t})$. The logits, or the log probabilities before normalization, are indicated by $z(y^k|x, y1:k-1)$, linking directly to the model's raw outputs before they are passed through the softmax function $\pi_\theta(y^{t+1}|x, y^{1:t}) = \text{softmax}(z(y^{t+1}|x, y^{1:t}))$. The weight matrix of the model's output layer is denoted by $W$, and the hidden state of the model at the $k$-th position given context $[x, y^{1:k-1}]$ is represented by $h_L(x, y^{1:k-1})$. The logit of the token $y$ is defined as $z(y|x, y^{1:k-1}) = \hat{y}^\top W h_L(x, y^{1:k-1})$, where $\hat{y}$ is the one-hot vector corresponding to the token $y$. The reward function $r(x, y) = \beta \log \frac{\pi_\theta(y|x)}{\pi_{\text{ref}}(y|x)}$ is used to calculate the reward signal for the model's outputs.

The loss function of DPO is defined as follows:

$$\mathcal{L}_{\text{DPO}}(y_w, y_l) = \log \frac{\exp(\beta \log \frac{\pi_\theta(y_w|x)}{\pi_{\text{ref}}(y_w|x)})}{\exp(\beta \log \frac{\pi_\theta(y_w|x)}{\pi_{\text{ref}}(y_w|x)}) + \exp(\beta \log \frac{\pi_\theta(y_l|x)}{\pi_{\text{ref}}(y_l|x)})} \quad (5)$$

where $y_w$ and $y_l$ are the correct and incorrect responses, respectively, $\pi_\theta(y|x)$ is the model's predicted probability of generating response $y$ given input $x$, and $\pi_{\text{ref}}(y|x)$ is the reference model's

---

[1]https://huggingface.co/datasets/xinlai/Math-Step-DPO-10K

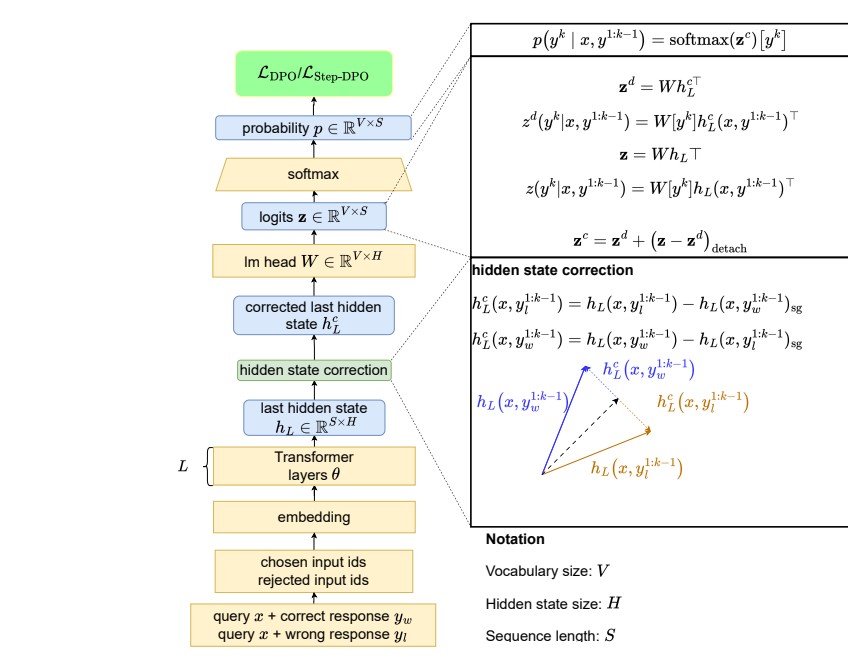

Figure 1: Illustration of the FDT algorithm. The model's hidden states are decomposed into shared semantic components and distinctive semantic components. FDT focuses on the distinctive semantic components to enhance the model's ability to distinguish between correct and incorrect mathematical responses.

predicted probability of generating response $y$ given input $x$. The gradient of the DPO loss function with respect to the logits of the model is given by:

$$
\begin{cases}
\frac{\partial \mathcal{L}_{\text{DPO}}(y_w, y_l)}{\partial z(y|x, y_w^{1:k-1})} & = \beta \frac{\exp(r(x,y_l))}{\exp(r(x,y_w)) + \exp(r(x,y_l))} \left( \mathbb{I}_{y=y_w^k} - \text{softmax}(z(y|x, y_w^{1:k-1})) \right) \\
\frac{\partial \mathcal{L}_{\text{DPO}}(y_w, y_l)}{\partial z(y|x, y_l^{1:k-1})} & = -\beta \frac{\exp(r(x,y_l))}{\exp(r(x,y_w)) + \exp(r(x,y_l))} \left( \mathbb{I}_{y=y_l^k} - \text{softmax}(z(y|x, y_l^{1:k-1})) \right)
\end{cases}
\tag{6}
$$

We denote $c(x, y_w, y) = \frac{\partial \mathcal{L}(y_w, y_l)}{\partial z(y|x, y_w^{1:k-1})}$ and $c(x, y_l, y) = \frac{\partial \mathcal{L}(y_w, y_l)}{\partial z(y|x, y_l^{1:k-1})}$.

The gradient of $y$-th row in the weight matrix $W[y] = y^\top W$ is

$$
\begin{aligned}
\frac{\partial \mathcal{L}_{\text{DPO}}(y_w, y_l)}{\partial W[y]} &= c(x, y_w, y) \frac{\partial z(y|x, y_w^{1:k-1})}{\partial W[y]} + c(x, y_l, y) \frac{\partial z(y|x, y_l^{1:k-1})}{\partial W[y]} \\
&= c(x, y_w, y) \frac{\partial W[y] h(x, y_w^{1:k-1})}{\partial W[y]} + c(x, y_l, y) \frac{\partial W[y] h(x, y_l^{1:k-1})}{\partial W[y]} \\
&= c(x, y_w, y) h(x, y_w^{1:k-1})^\top + c(x, y_l, y) h(x, y_l^{1:k-1})^\top \\
&= c(x, y_w, y) h(x, y_w^{1:k-1})^\top + c(x, y_l, y) h(x, y_l^{1:k-1})^\top,
\end{aligned}
\tag{7}
$$

where $h(x, y^{1:k-1})$ is the hidden state of the model at the $k$-th position given context $[x, y^{1:k-1}]$. If we use gradient descent to update the weight matrix $W$, the update of $W$ is a covex combination of the hidden states $h(x, y_w^{1:k-1})$ and $h(x, y_l^{1:k-1})$.

However, hidden states within the embedding space exhibit a high concentration of semantic information. For any pair of correct and incorrect responses $h_L(x, y_w^{1:k-1})$, $h_L(x, y_l^{k-1})$, these states can be decomposed into two components: a shared semantic component $h_s = \frac{1}{2}\left(h_L(x, y_w^{1:k-1}) + h_L(x, y_l^{k-1})\right)$ and a distinctive semantic component $h_d = h_L(x, y_w^{1:k-1}) - h_L(x, y_l^{k-1})$.

The shared semantic component $h_s$ encompasses semantic features shared by both responses, such as surface characteristics, contributing to their similarity. In contrast, the distinctive semantic component $h_d$ contains the semantic features crucial for distinguishing between the correct and incorrect responses. Therefore, focusing on the differential component of the hidden states can significantly enhance the model's performance in mathematical reasoning tasks, as it directs attention to the semantic distinctions critical for accuracy **?**.

In order to focus on the differences between correct and incorrect responses, we hope to correct the update of $W$ to amplify the hidden states that contribute more to the differences. The update of $W$ is corrected as follows:

$$\Delta W[y] = \alpha c(x, y_w, y)(\underbrace{h(x, y_w^{1:k-1}) - h(x, y_l^{1:k-1})})^\top + \alpha c(x, y_l, y)(\underbrace{h(x, y_l^{1:k-1}) - h(x, y_w^{1:k-1})})^\top$$

$$= \alpha(c(x, y_w, y) - c(x, y_l, y))(h(x, y_w^{1:k-1}) - h(x, y_l^{1:k-1}))^\top,$$

(8)

where $\alpha$ is the learning rate.

## 4.1 FDT ALGORITHM

In this section, we introduce the FDT algorithm, which corrects the update of the model head weight matrix $W$ to amplify the hidden states that contribute more to the differences between correct and incorrect responses according to Equation 8. The FDT algorithm is shown in the Figure 1. The algorithm consists of 5 steps.

**Extraction of Hidden States**   The FDT process begins by extracting the hidden states from the last transformer layer of the language model for both the correct and incorrect responses. These states are denoted as $h_L(x, y_w^{1:k-1})$ and $h_L(x, y_l^{1:k-1})$, respectively. Concurrently, we also extract the logits associated with both the correct and incorrect responses, collectively represented as $\mathbf{z} = W h_L$, where $h_L = [h_L(x, y_w^{1:k-1})^\top, h_L(x, y_l^{1:k-1})^\top]^\top$.

**Computation of Differential Hidden State**   To emphasize the discrepancies between the correct and incorrect reasoning processes within the model, we compute the differential hidden state. This is achieved by subtracting the hidden state corresponding to the incorrect response from that of the correct response: $h_L^c(x, y_w^{1:k-1}) = h(x, y_w^{1:k-1}) - h(x, y_l^{1:k-1})_{sg}$ and $h_L^c(x, y_l^{1:k-1}) = h(x, y_l^{1:k-1}) - h(x, y_w^{1:k-1})_{sg}$, where the subscript $sg$ denotes that the gradient is not backpropagated. This differential hidden state encapsulates the critical differences that the model needs to learn in order to discern between correct and incorrect mathematical reasoning.

**Recomputation of Logits**   Utilizing the differential hidden state $h_L^c$, we recompute the logits $\mathbf{z}_d$ that specifically reflect the semantic distinctions critical for accurate response generation: $\mathbf{z}_d = W h_L^c$, where $h_L^c = [h_L^c(x, y_w^{1:k-1})^\top, h_L^c(x, y_l^{1:k-1})^\top]^\top$.

**Correction of Logits**   To integrate the newly computed differential logits with the original logits while preserving the model's ability to perform general reasoning, we compute the corrected logits $\mathbf{z}_c$. This is performed by blending the differential logits with the original logits, where the original logits are detached from the gradient computation to make sure the weight is only updated with the differential hidden state: $\mathbf{z}_c = \mathbf{z}_d + (\mathbf{z} - \mathbf{z}_d)_{sg}$.

**Compute Loss Fuction**   We first compute the log probability of the correct response and the incorrect response: $\log \pi(y|x, y_w^{1:k-1}) = z_c(y|x, y_w^{1:k-1}) - \log \sum_{y'} \exp(z_c(y'|x, y_w^{1:k-1}))$ and $\log \pi(y|x, y_l^{1:k-1}) = z_c(y|x, y_l^{1:k-1}) - \log \sum_{y'} \exp(z_c(y'|x, y_l^{1:k-1}))$. Then, we compute the loss function. As the FDT is a plug-in algorithm, the loss function can be any loss function with pair-wise samples as input. The the parameters of the model are updated by the gradient of the loss function.

This operation ensures that the corrections made by FDT are grounded in the model's initial predictions, thereby facilitating a refined adjustment that enhances the model's accuracy in distinguishing correct from incorrect responses without losing the contextual grounding provided by the original logits.

---

**Algorithm 1** Focused Differentiation Training (FDT)

---

**Input:** Query $x$, reference sequence $y_w$, label sequence $y_l$, learning rate $\alpha$, number of iterations $K$

**for** $n = 1$ to $N$ **do**

Compute the last layer hidden states of the model $h_L(x, y_w^{1:k-1})$ and $h_L(x, y_l^{1:k-1})$ for $k = 1, 2, \ldots, K$

Compute the logits $\mathbf{z}_l$ and $\mathbf{z}_w$ given context $[x, y_l^{1:k-1}]$ and $[x, y_w^{1:k-1}]$, $\mathbf{z}_l = W h_L(x, y_l^{1:k-1})$ and $\mathbf{z}_w = W h_L(x, y_w^{1:k-1})$

Compute the differential last layer hidden state

$$h_L^c(x, y_l^{1:k-1}) = h_L(x, y_l^{1:k-1}) - h_L(x, y_w^{1:k-1})$$

and

$$h_L^c(x, y_w^{1:k-1}) = h_L(x, y_w^{1:k-1}) - h_L(x, y_l^{1:k-1})$$

Compute the differential logits $\mathbf{z}_l^d = W h_L^c(x, y_l^{1:k-1})$ and $\mathbf{z}_w^d = W h_L^c(x, y_w^{1:k-1})$

Compute the corrected logits $\mathbf{z}_w^c = \mathbf{z}_w^d + (\mathbf{z}_w - \mathbf{z}_w^d)_{sg}$ and $\mathbf{z}_l^c = \mathbf{z}_l^d + (\mathbf{z}_l - \mathbf{z}_l^d)_{sg}$

Compute the log probabilities of the tokens $y_w^k$ and $y_l^k$ given context $[x, y_w^{1:k-1}]$ and $[x, y_l^{1:k-1}]$,

$$\log \pi_\theta(y_w^k | x, y_w^{1:k-1}) = \log \text{softmax}(z_w^c(y_w^k | x, y_w^{1:k-1}))$$

and

$$\log \pi_\theta(y_l^k | x, y_l^{1:k-1}) = \log \text{softmax}(z_l^c(y_l^k | x, y_l^{1:k-1}))$$

.

Compute the loss

$$\mathcal{L}(y_w, y_l) = \log \frac{\exp(\beta \log \frac{\pi_\theta(y_w | x)}{\pi_{\text{ref}}(y_w | x)})}{\exp(\beta \log \frac{\pi_\theta(y_w | x)}{\pi_{\text{ref}}(y_w | x)}) + \exp(\beta \log \frac{\pi_\theta(y_l | x)}{\pi_{\text{ref}}(y_l | x)})}$$

.

Update the model weight using the gradients.

**end for**

---

The FDT algorithm is shown in Algorithm 1. We can prove that the FDT algorithm can be used to correct the update of the model head weight matrix $W$ by the differences between correct and incorrect responses.

**Theorem 1.** *The FDT algorithm can be used to correct the update of the model head weight matrix $W$ as illustrated in Equation 8.*

Given that the shared semantic component of the hidden states does not contribute to the correctness of the response, we anticipate that its influence on the logits will be minimal following the update of the model's output layer weight matrix $W$.

The Theorem 2 shows that FDT can effectively control the influence of the shared semantic component of the hidden states of the model on the logits after the update of the model head weight matrix $W$. Prior to detailing this theorem, we shall first define the concepts of the $\eta$-subexponential distribution and the $\eta$-subexponential vector, which are instrumental in understanding the underlying mechanisms of our approach.

A random variable $X$ is defined as $\eta$-subexponential ($\eta$-subE) for $\eta \in (0, 2)$ if its $\eta$-norm, $\|X\|_{\psi_\eta}$, determined by $\|X\|_{\psi_\eta} = \inf\{t > 0 : \mathbb{E}\exp((|X|/t)^\eta) \leq 2\}$, is finite. We define a vector $Y$ as an $\eta$-subE vector with mean $\mu$, covariance $\Sigma$, and a norm upper bound $K$, if the transformed vector $\Sigma^{-1/2}(Y - \mu)$ has components that are $\eta$-subE with unit variance and are bounded by $K$. Furthermore, we denote $D_Y \sim \mathcal{E}_\eta(\mu, \Sigma, K)$ to indicate that $D_Y$ comprises independent and identically distributed (i.i.d.) samples drawn from an $\eta$-subE distribution for vectors characterized by mean $\mu$, covariance $\Sigma$, and norm bound $K$.

The hidden states from correct responses are modeled as $D_+ \sim \mathcal{E}_\eta(\mu_+, \Sigma_+, K)$, and the non-preferred hidden states from incorrect responses as $D_- \sim \mathcal{E}_\eta(\mu_-, \Sigma_-, K)$. This modeling is rea-

sonable as the $\alpha$-subexponential distribution is a general distribution includes any sub-Gaussian distribution as well as any sub-exponential distribution such as normal or $\chi^2$ distributions and allows for heavier tails. This modeling is also adopted in the previous work **?**.

**Theorem 2.** *Assume that* $\|\mu_+\|^2 - \|\mu_-\|^2 \leq \delta$, *and the hidden states are bounded* $\|\mu_+\|^2 \leq M$ *and* $\|\mu_-\|^2 \leq M$. $\|\Sigma_+ + \Sigma_-\| < c_v\sqrt{d}$. *The update of the model head weight matrix* $\Delta W$ *satisfies*

$$\Delta W[y](h(y|x, y_w^{1:k-1}) + h(y|x, y_l^{1:k-1})) \leq 4\alpha\delta, \tag{9}$$

*with probability at least* $1 - 2\exp\left(-\frac{\delta^\eta}{2^{(\eta+1)}M^\eta c_v\sqrt{d}}\right) - 2\exp\left(-\frac{M^\eta}{2c_v\sqrt{d}}\right)$.

The proof of Theorem 2 is deferred to the appendix A.2. We also show that the FDT algorithm can emphasize the distinctive features of correct responses over incorrect ones.

**Corollary 1.** *Assume that* $\|\Sigma_+ + \Sigma_-\| < c_v\sqrt{d}$. *The update of the model head weight matrix* $\Delta W$ *satisfies*

$$\Delta W[y](h(y|x, y_w^{1:k-1}) - h(y|x, y_l^{1:k-1})) \geq \frac{1}{2}\alpha\|\mu_+ - \mu_-\|, \tag{10}$$

*with probability at least* $1 - 2\exp\left(-\frac{\|\mu_+ - \mu_-\|^\eta}{2^{\eta+1}c_v\sqrt{d}}\right)$.

The proof of Corollary 1 is deferred to the appendix A.3.

**Remark 1.** *Theorem 2 and Corrolary 1 show that the shared semantic component of the hidden states of the model has a limited influence on the logits after the update of the model head weight matrix $W$. This limited influence is crucial in ensuring that the adjustments made to the weight matrix $W$ effectively mitigate the potential overgeneralization brought about by the shared semantic component, while focusing on the distinctive features of responses.*

We also provide an empirical evidence to support these theoretical results. Figure 2 shows reward margins between the DPO and FDT algorithms in the Figure 2. The reward margin is defined as $\beta\log\frac{\pi_\theta(y_w|x)}{\pi_{\text{ref}}(y_w|x)} - \beta\log\frac{\pi_\theta(y_l|x)}{\pi_{\text{ref}}(y_l|x)}$. Figure 2 shows that FDT leads to a larger margin between the logits of the correct response and the incorrect response. The results demonstrate that the FDT algorithm effectively enhances the model's ability to differentiate between correct and incorrect responses, thereby improving the model's performance in mathematical reasoning tasks.

## 5 EXPERIMENTS

### 5.1 DATASETS

In our supervised fine-tuning phase, we utilize the NuminaMath-Co portion of the metamath-qwen2-math dataset. During the DPO/Step-DPO stages, we incorporate datasets from Step-DPO which consist of 10,000 pairwise preference data points. For assessing performance, we employ the widely recognized datasets: MATH **?**, GSM8K **?**, and MMLU-redux **?**, using accuracy as our primary metric for evaluation. The MATH dataset includes 5000 mathematical problems across five levels of difficulty and seven categories, such as algebra, counting and probability, geometry, intermediate algebra, number theory, prealgebra, and precalculus. The GSM8K dataset comprises 1319 mathematical problems, each accompanied by step-by-step solutions and verified answers, typically presenting less complexity than those found in the MATH dataset. We also use the MMLU-redux dataset, which contains 3000 questions across a diverse range of subjects to assesses both the breadth and depth of language understanding capabilities of the model.

### 5.2 BASELINES

We compare our proposed method with the following baselines: DPO **?** and Step-DPO **?**. We evaluate the performance of our method against these baselines on several models, including Llama-3.2-3B-Instruct and Qwen2.5-3B-Instruct.

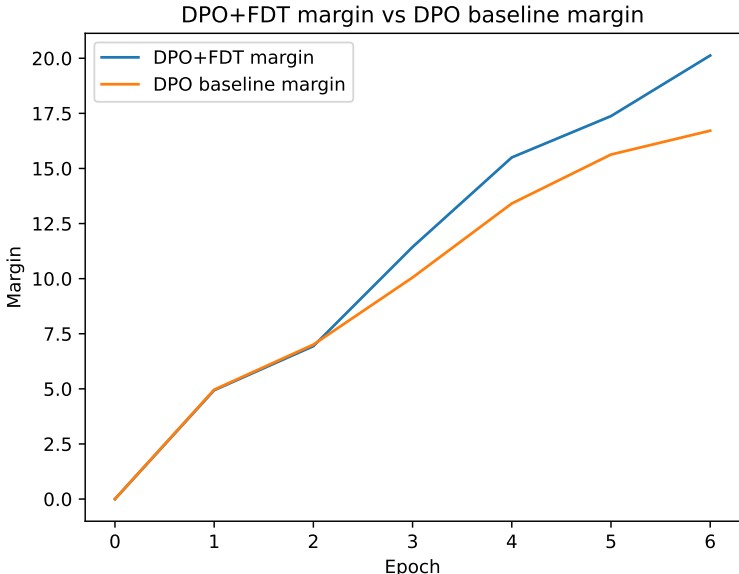

Figure 2: The reward margin between the DPO and FDT algorithms. We conduct this experiment based on Qwen2.5-3B-Instruct model and the Math-Step-DPO-10K dataset under the same setting.

### 5.3 IMPLEMENTATION DETAILS

For the Llama-3.2-3B-Instruct and Qwen2.5-3B-Instruct models, we initially conduct supervised fine-tuning using the NuminaMath-Co dataset. This stage employs the AdamW optimizer paired with a linear decay learning rate scheduler. We establish a warmup ratio of 0.03, a global batch size of 256, and set the learning rate at $5 \times 10^{-6}$.

Subsequent to fine-tuning, we implement the Direct Preference Optimization (DPO) and Step-DPO processes. Both the DPO and Step-DPO baselines are configured with a learning rate of $5 \times 10^{-6}$ and a training duration of 8 epochs. For our proposed method under these stages, the learning rate is slightly increased to $7 \times 10^{-6}$, while maintaining the same duration and batch size. All models during these stages utilize the AdamW optimizer, with a cosine learning rate schedule and warmup ratio of 0.1. The hyperparameter $\beta$ is set to 0.4 for DPO and Step-DPO processes without FDT, and 0.5 for DPO and Step-DPO processes with FDT.

### 5.4 RESULTS

The performance results of our FDT method compared to the established baselines on the GSM8K, MATH, and MMLU datasets are summarized in Table 1. Across these diverse datasets, FDT not only meets but often exceeds the performance metrics of the baseline models. This consistent outperformance across all models and datasets underscores the efficacy of FDT in enhancing mathematical reasoning capabilities of language models. Specifically, the FDT method has marked a significant improvement in the performance of the Qwen2.5-3B-Instruct model. On the GSM8K dataset, it achieved an accuracy of 79.7%, surpassing the baseline by 2.4 percentage points. Similarly, on the MATH dataset, FDT recorded a substantial increase in accuracy, reaching 61.5%, which represents an enhancement of 4.3 percentage points over the baseline. On the MMLU-redux dataset, the method managed to achieve a notable accuracy of 63.6% with an improvement of 1.0 percentage point. In applying our method to Llama-3.2-3B-Instruct, we also observed performance enhancements, which corroborates the robustness and generalizability of our approach. The improvement in the MMLU-redux dataset shows that model can benefit from the FDT method in a broader range of tasks beyond mathematical reasoning.

Table 1: The performance of the models on the GSM8K, MATH, and MMLU-redux datasets.

| model | GSM8K | MATH | MMLU-redux |
|---|---|---|---|
| Llama-3.2-3B-Instruct+SFT | 43.1 | 33.3 | 0.47 |
| Llama-3.2-3B-Instruct+DPO | 46.6 | 32.9 | 1.12 |
| Llama-3.2-3B-Instruct+DPO+FDT | **47.5** | 33.2 | **7.48** |
| Llama-3.2-3B-Instruct+Step-DPO | 46.9 | 33.2 | 1.04 |
| Llama-3.2-3B-Instruct+Step-DPO+FDT | 46.9 | **34.0** | 0.76 |
| Qwen2.5-3B-Instruct+SFT | 77.3 | 56.9 | 60.5 |
| Qwen2.5-3B-Instruct+DPO | 77.5 | 47.2 | 62.6 |
| Qwen2.5-3B-Instruct+DPO+FDT | 77.6 | **61.5** | **63.6** |
| Qwen2.5-3B-Instruct+Step-DPO | 77.3 | 55.4 | 62.1 |
| Qwen2.5-3B-Instruct+Step-DPO+FDT | **79.7** | 58.6 | 62.2 |

## 6 CONCLUSION

In this paper, we presented FDT, an innovative training methodology developed to bolster the mathematical reasoning capabilities of LLMs. FDT harnesses the dense semantic information embedded in the hidden states of LLMs to effectively discriminate between correct and incorrect mathematical responses, particularly in complex multi-step scenarios. Unlike other methodologies that depend on human intervention or more sophisticated models for response evaluation at the token or logits level, our approach autonomously enhances mathematical reasoning by capitalizing on the inherent capabilities of the model itself. This self-sufficient method improves performance without external aid, making full use of the model's native capacities. Our empirical evaluations on the GSM8K, MATH, and MMLU-redux datasets have demonstrated that FDT not only significantly enhances accuracy but also deepens reasoning capabilities. It notably excels in providing detailed feedback on partially correct solutions, a feature particularly valuable in educational settings. Additionally, FDT's seamless integration with existing RLHF frameworks underscores its versatility.

## A PROOF

### A.1 PROOF OF THEOREM 1

*Proof.* We denote the hidden states of the model as $h(x, y^{1:k-1})$ and $\log \pi(y|x) = z(x, y) - \log \sum_{y'} \exp(z(x, y'))$. The loss function of FDT is

$$\mathcal{L}(y_w, y_l) = \log \frac{\exp(\beta \log \frac{\pi_\theta(y_w|x)}{\pi_{\text{ref}}(y_w|x)})}{\exp(\beta \log \frac{\pi_\theta(y_w|x)}{\pi_{\text{ref}}(y_w|x)}) + \exp(\beta \log \frac{\pi_\theta(y_l|x)}{\pi_{\text{ref}}(y_l|x)})} \tag{11}$$

The gradient of the loss function of FDT can be derived by the chain rule as

$$\begin{aligned}
\frac{\partial}{\partial z(y|x, y_w^{1:k-1})} \mathcal{L}(y_w, y_l) &= \frac{\partial \mathcal{L}(y_w, y_l)}{\partial r(x, y_w)} \frac{\partial r(x, y_w)}{\partial z(y|x, y_w^{1:k-1})} \\
&= \frac{\partial \mathcal{L}(y_w, y_l)}{\partial r(x, y_w)} \frac{\partial r(x, y_w)}{\partial z_w^c(y|x, y_w^{1:k-1})} \frac{\partial z_w^c(y|x, y_w^{1:k-1})}{\partial z(y|x, y_w^{1:k-1})} \\
&= c(x, y_w, y) \frac{\partial z_w^c(y|x, y_w^{1:k-1})}{\partial z(y|x, y_w^{1:k-1})} \\
&= c(x, y_w, y)
\end{aligned} \tag{12}$$

Similarly, we can derive the gradient of the loss function of FDT with respect to $z(y|x, y_l^{1:k-1})$ as

$$\frac{\partial}{\partial z(y|x, y_l^{1:k-1})} \mathcal{L}(y_w, y_l) = c(x, y_l, y) \tag{13}$$

The gradient of the loss function of FDT with respect to the weight matrix $W$ is

$$\Delta W[y] = \alpha \frac{\partial}{\partial W[y]} \mathcal{L}(y_w, y_l) = \alpha \frac{\partial \mathcal{L}(y_w, y_l)}{\partial z_w^c(y|x, y_w^{1:k-1})} (h(x, y_w^{1:k-1}) - h(x, y_l^{1:k-1}))^\top$$

$$+ \alpha \frac{\partial \mathcal{L}(y_w, y_l)}{\partial z_l^c(y|x, y_l^{1:k-1})} (h(x, y_l^{1:k-1}) - h(x, y_w^{1:k-1}))^\top \tag{14}$$

$$= \alpha c(x, y_w, y)(h(x, y_w^{1:k-1}) - h(x, y_l^{1:k-1}))^\top$$

$$+ \alpha c(x, y_l, y)(h(x, y_l^{1:k-1}) - h(x, y_w^{1:k-1}))^\top$$

$$= \alpha (c(x, y_w, y) - c(x, y_l, y))(h(x, y_w^{1:k-1}) - h(x, y_l^{1:k-1}))^\top,$$

which is consistent with Equation 8. $\qquad\square$

### A.2 PROOF OF THEOREM 2

*Proof.* We assume that the hidden states of correct response and incorrect response is similar, $\|\mu_+\|^2 - \|\mu_-\|^2 \leq \delta^2$, and the hidden states are bounded $\|\mu_+\|^2 \leq M$ and $\|\mu_-\|^2 \leq M$. $\|\Sigma_+ + \Sigma_-\| < c_v\sqrt{d}$.

$$\Delta W[y] = \alpha \frac{1}{n} \sum_{i=1}^{n} [(c(x_i, y_{wi}, y) - c(x_i, y_{li}, y))(h(x, y_{wi}^{1:k-1}) - h(x, y_{li}^{1:k-1}))]^\top, \tag{15}$$

where $n$ is the number of samples.

$$\left(h(y|x, y_w^{1:k-1}) - h(y|x, y_l^{1:k-1})\right)^\top \left(h(y|x, y_w^{1:k-1}) + h(y|x, y_l^{1:k-1})\right)$$

$$= \left(h(y|x, y_w^{1:k-1}) - h(y|x, y_l^{1:k-1})\right)^\top \left(h(y|x, y_w^{1:k-1}) + h(y|x, y_l^{1:k-1}) - \mu_+ - \mu_-\right)$$

$$+ \left(h(y|x, y_w^{1:k-1}) - h(y|x, y_l^{1:k-1})\right)^\top (\mu_+ + \mu_-)$$

$$= \left(h(y|x, y_w^{1:k-1}) - h(y|x, y_l^{1:k-1})\right)^\top \left(h(y|x, y_w^{1:k-1}) + h(y|x, y_l^{1:k-1}) - \mu_+ - \mu_-\right)$$

$$+ \left((\mu_+ - \mu_-)^\top(\mu_+ + \mu_-) + \left(h(y|x, y_w^{1:k-1}) - h(y|x, y_l^{1:k-1}) - \mu_+ - \mu_-\right)^\top (\mu_+ + \mu_-)\right)$$

$$= (\mu_+ - \mu_-)^\top (h(y|x, y_w^{1:k-1}) + h(y|x, y_l^{1:k-1}) - \mu_+ - \mu_-)$$

$$+ \left(h(y|x, y_w^{1:k-1}) - h(y|x, y_l^{1:k-1}) - \mu_+ + \mu_-\right)^\top (h(y|x, y_w^{1:k-1}) + h(y|x, y_l^{1:k-1}) - \mu_+ - \mu_-)$$

$$+ \left((\mu_+ - \mu_-)^\top(\mu_+ + \mu_-) + \left(h(y|x, y_w^{1:k-1}) - h(y|x, y_l^{1:k-1}) - \mu_+ - \mu_-\right)^\top (\mu_+ + \mu_-)\right) \tag{16}$$

The difference hidden state between the hidden states of the model from correct and incorrect response is an $\eta$-subexponential vector, which is drawn from the $\eta$-subexponential distribution $\mathcal{E}_\eta(\mu_+ - \mu_-, \Sigma_+ + \Sigma_-, K)$ and the common hidden state is also an $\eta$-subexponential vector, which is drawn from the $\eta$-subexponential distribution $\mathcal{E}_\eta(\mu_+ + \mu_-, \Sigma_+ + \Sigma_-, K)$.

Therefore, we have

$$P(\|h(y|x, y_w^{1:k-1}) + h(y|x, y_l^{1:k-1}) - \mu_+ - \mu_-\| \geq t)$$

$$= P(|(h(y|x, y_w^{1:k-1}) + h(y|x, y_l^{1:k-1}) - \mu_+ - \mu_-)^\top a| \geq t)$$

$$\leq 2\exp\left(-\frac{t^\eta}{2a^\top(\Sigma_+ + \Sigma_-)a}\right)$$

$$\leq 2\exp\left(-\frac{t^\eta}{2c_v\sqrt{d}}\right),$$

for any unit vector $a$. If we select $a = \frac{\mu_+ + \mu_-}{\|\mu_+ + \mu_-\|}$,

$$P((\mu_+ + \mu_-)^\top(h(y|x, y_w^{1:k-1}) + h(y|x, y_l^{1:k-1}) - \mu_+ - \mu_-) \geq 2Mt) \leq 2\exp\left(-\frac{t^\eta}{2c_v\sqrt{d}}\right)$$

With probability $p_1 = 1 - 2\exp\left(-\frac{\delta^\eta}{2^{(\eta+1)}M^\eta c_v \sqrt{d}}\right)$, we have

$$(\mu_+ + \mu_-)^\top (h(y|x, y_w^{1:k-1}) + h(y|x, y_l^{1:k-1}) - \mu_+ - \mu_-) \le \delta.$$

Similarly, we have

$$\left(\mu_+ - \mu_-\right)^\top (h(y|x, y_w^{1:k-1}) + h(y|x, y_l^{1:k-1}) - \mu_+ - \mu_-) \le \delta$$

with probability at least $p_1$.

$$\left(h(y|x, y_w^{1:k-1}) - h(y|x, y_l^{1:k-1}) - \mu_+ + \mu_-\right)^\top (h(y|x, y_w^{1:k-1}) + h(y|x, y_l^{1:k-1}) - \mu_+ - \mu_-)$$

$$\le \|h(y|x, y_w^{1:k-1}) - h(y|x, y_l^{1:k-1}) - \mu_+ + \mu_-\| \|h(y|x, y_w^{1:k-1}) + h(y|x, y_l^{1:k-1}) - \mu_+ - \mu_-\|$$

$$\le \|h(y|x, y_w^{1:k-1}) - h(y|x, y_l^{1:k-1}) - \mu_+ + \mu_-\|\frac{\delta}{M} \le M\frac{\delta}{M} = \delta$$

with probability $p_2 = p_1\left(1 - 2\exp\left(-\frac{M^\eta}{2c_v\sqrt{d}}\right)\right) \ge 1 - 2\exp\left(-\frac{\delta^\eta}{2^{(\eta+1)}M^\eta c_v\sqrt{d}}\right) - 2\exp\left(-\frac{M^\eta}{2c_v\sqrt{d}}\right)$.

With probability at least $1 - 2\exp\left(-\frac{\delta^\eta}{2^{(\eta+1)}M^\eta c_v\sqrt{d}}\right) - 2\exp\left(-\frac{M^\eta}{2c_v\sqrt{d}}\right)$,

$$\left(h(y|x, y_w^{1:k-1}) - h(y|x, y_l^{1:k-1})\right)^\top \left(h(y|x, y_w^{1:k-1}) + h(y|x, y_l^{1:k-1})\right) \le 4\delta.$$

Therefore, we have $\Delta W[y](h(y|x, y_w^{1:k-1}) + h(y|x, y_l^{1:k-1})) \le 5\alpha\delta(c(x, y_w, y) - c(x, y_l, y))$ with probability at least

$$1 - 2\exp\left(-\frac{\delta^\eta}{2^{(\eta+1)}M^\eta c_v\sqrt{d}}\right) - 2\exp\left(-\frac{M^\eta}{2c_v\sqrt{d}}\right)$$

.

$\square$

### A.3 PROOF OF CORROLARY 1

*Proof.* With probability at least $1 - 2\exp\left(-\frac{\|\mu_+ - \mu_-\|^\eta}{2^{\eta+1}c_v\sqrt{d}}\right)$,

$$\left\|h(y|x, y_w^{1:k-1}) - h(y|x, y_l^{1:k-1}) - \mu_+ + \mu_-\right\| \le \frac{1}{2}\|\mu_+ - \mu_-\|$$

$\square$

From triangle inequality, we have

$$\left\|h(y|x, y_w^{1:k-1}) - h(y|x, y_l^{1:k-1})\right\| \ge \frac{1}{2}\|\mu_+ - \mu_-\|$$

holds with probability at least $1 - 2\exp\left(-\frac{\|\mu_+ - \mu_-\|^\eta}{2c_v\sqrt{d}}\right)$. Therefore, we have

$$W[y](h(y|x, y_w^{1:k-1}) - h(y|x, y_l^{1:k-1})) \ge \frac{1}{2}\alpha\|\mu_+ - \mu_-\|,$$

with probability at least $1 - 2\exp\left(-\frac{\|\mu_+ - \mu_-\|^\eta}{2^{\eta+1}c_v\sqrt{d}}\right)$.

## B  RELATIVE DIFFERENCE OF HIDDEN STATE

To investigate the relationship between chosen and rejected samples in the hidden state space, we analyzed the relative differences in their hidden state norms. Specifically, for each paired samples, we calculated the relative difference as the absolute difference between their hidden state norms divided by their average norms. Figure 3 illustrates the distribution of these relative differences across all sample pairs for two different models: Qwen2.5-3B-Instruct and Mistral-7B-Instruct-v0.3. The histograms reveal that the relative differences are predominantly concentrated around zero for both models. Qwen2.5-3B-Instruct exhibits a mean of 0.0321 and a median of 0.0220, while Mistral-7B-Instruct-v0.3 shows even smaller differences with a mean of 0.0100 and a median of 0.0081. These consistently small differences suggest that the samples maintain similar representation norms in the models' hidden state spaces, regardless of their chosen or rejected labels.

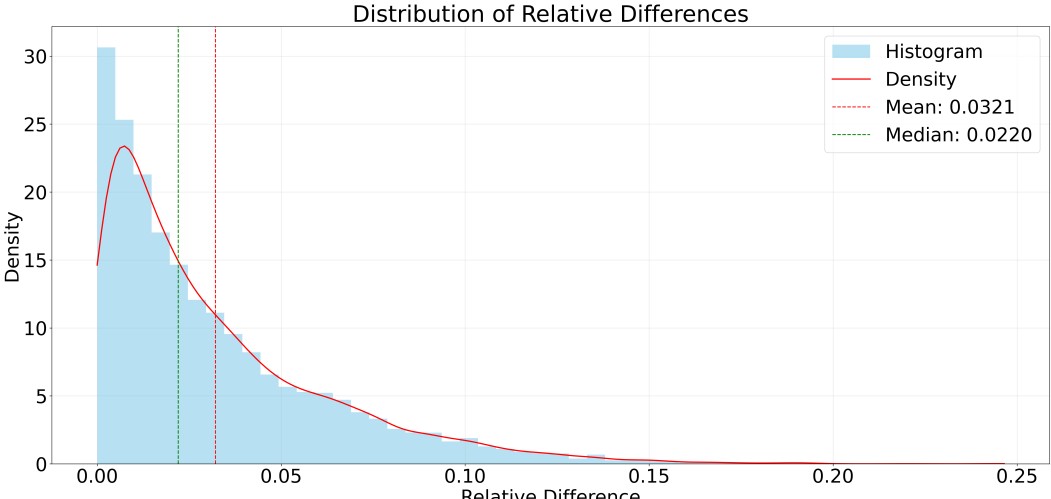

(a) Distribution of relative differences in hidden state norms of Qwen2.5-3B-Instruct between chosen and rejected samples.

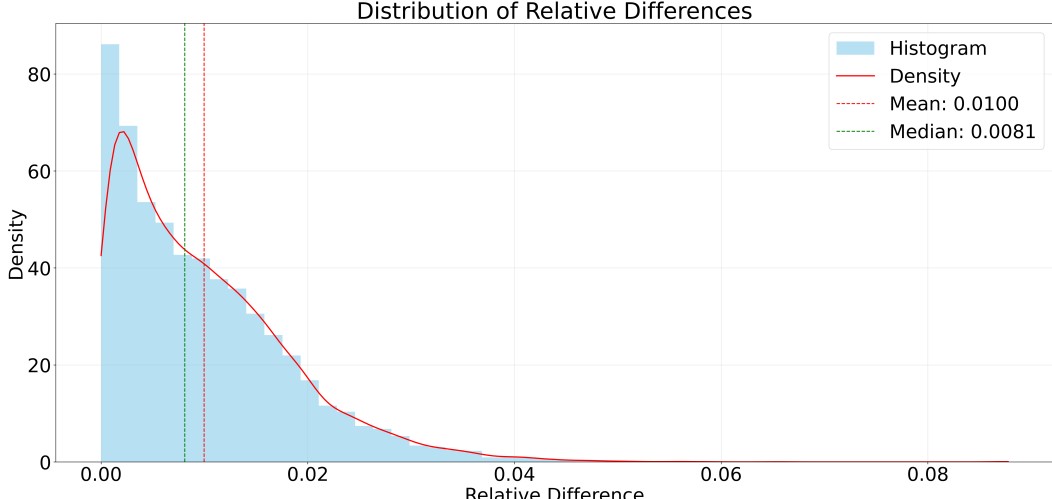

(b) Distribution of relative differences in hidden state norms of Mistral-7B-Instruct-v0.3 between chosen and rejected samples.

Figure 3: Distribution of relative differences in hidden state norms between chosen and rejected samples. The histogram shows that most differences are concentrated around zero, with a mean (red dashed line) and a median(green dashed line), indicating that paired samples maintain similar hidden state norms despite their different preference labels.

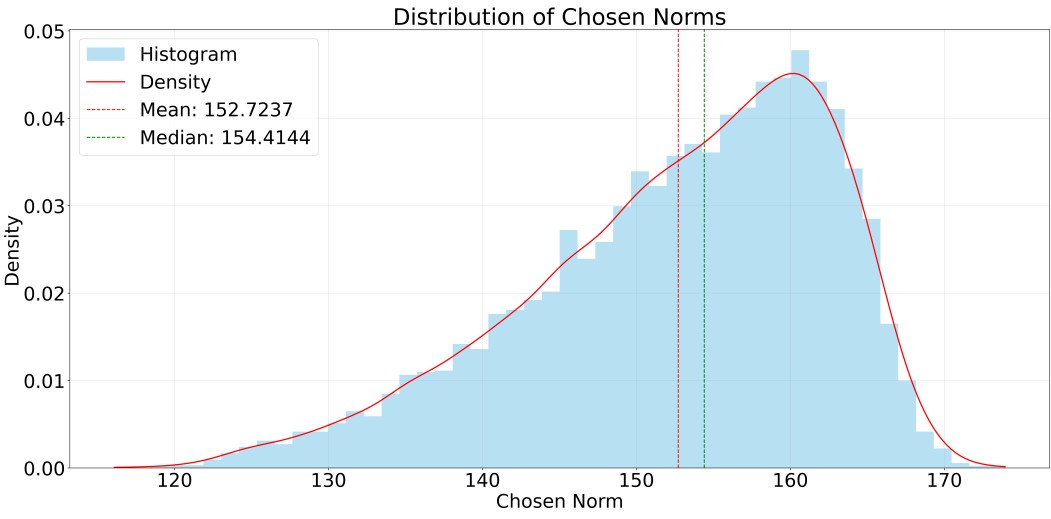

(a) The norm of chosen responses' hidden states.

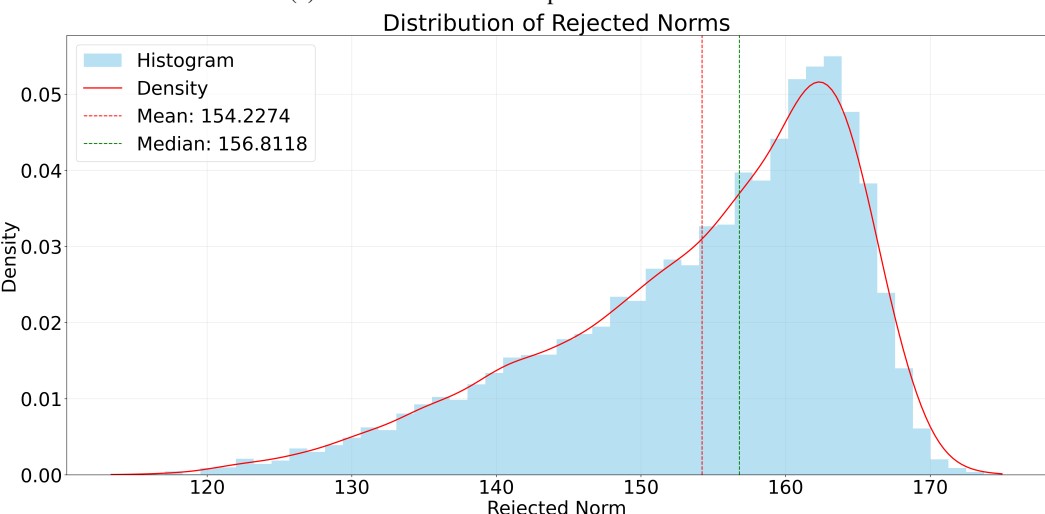

(b) The norm of rejected responses' hidden states.

Figure 4: Comparison for hidden states of Qwen2.5-3B-Instruct.

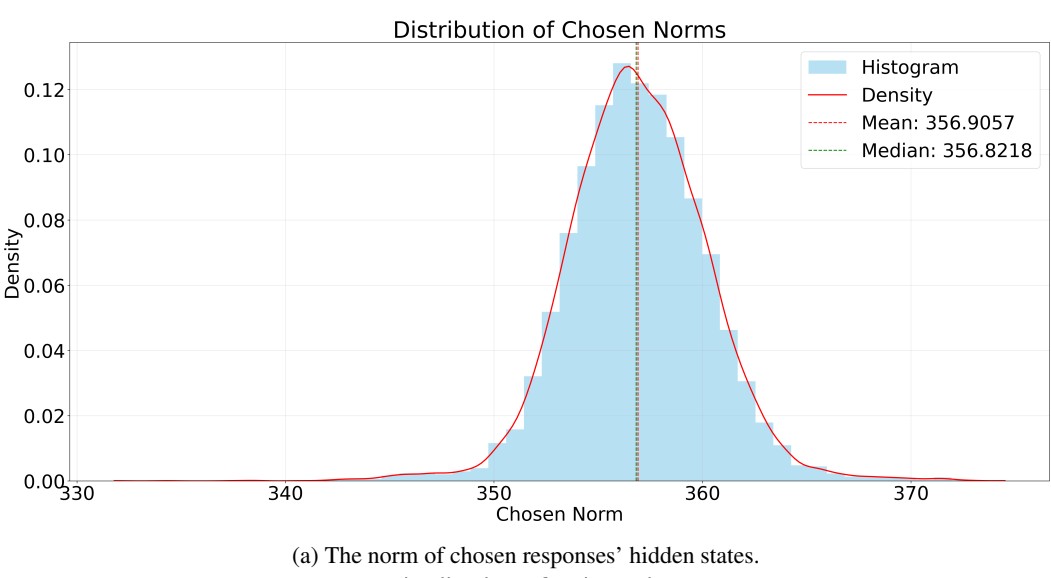

(a) The norm of chosen responses' hidden states.

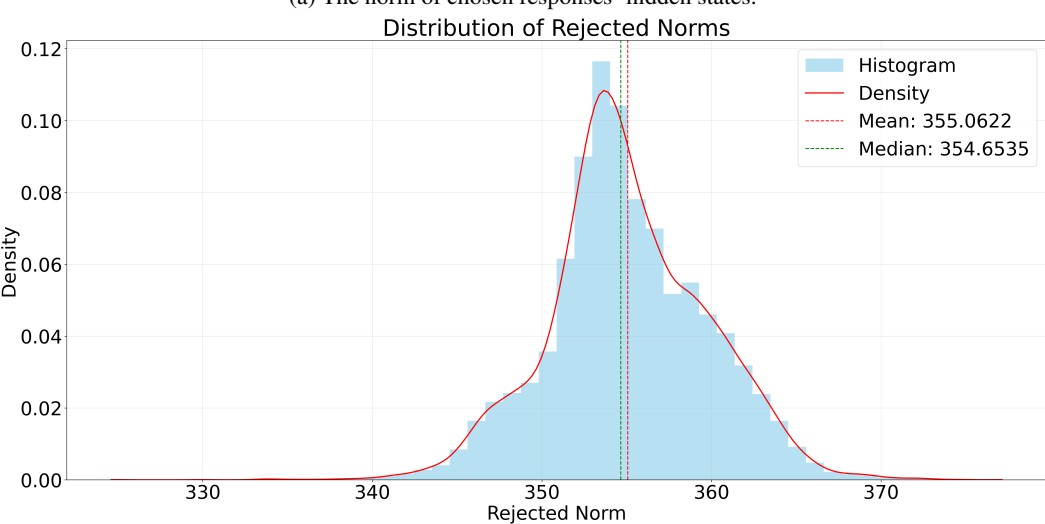

(b) The norm of rejected responses' hidden states.

Figure 5: Comparison for hidden states of Mistral-7B-Instruct-v0.3.

