# OpenReview forum: "Enhancing Mathematical Reasoning in Language Models Through Focused Differentiation Training"
_ICLR.cc/2025/Conference — Submitted to ICLR 2025_

### Official Review · Reviewer_b7S8 · 2024-10-27

**Soundness:** 4
**Presentation:** 4
**Contribution:** 3
**Rating:** 8
**Confidence:** 4

**Summary:**

This paper improves math reasoning by Focused Differentiation Training. That is, by comparing the win and lose answer, it calculates the difference between their embeddings. Then the algorithm takes gradients on the difference part, rather than the original embedding, using a stop gradient technique. By doing so, the model will get better signal from the main difference between the two solutions, rather than some common semantic features. This FDT algorithm shows good performance empirically on Math, GSM8K, and MMLU.

**Strengths:**

This paper proposes to take gradient only on the feature difference between the win and lose solutions, and shows this idea works well empirically.

- Originality: I am familiar with the math reasoning literature. As far as I know, this idea is novel and interesting. It makes sense to me, and can be potentially used by all other models, as demonstrated by Theorem 1.

- Quality: This is a nice paper presented with intuition, theorems, algorithms description, as well as experimental results. The theorems and intuitions are good and non-trivial, and the algorithm is also clearly stated, and easy to implement. I think this is a high quality paper.

- Clarity: this paper is very easy to follow, especially if the reader is familiar with DPO.

- Significance: I think this paper provides a very useful gadget for doing math reasoning. As stated in Theorem 1, it can be used for any other models as well. Therefore, I think many researchers in the field will be interested to try. However, since the improvements showed in the experiments are not very big, I will not say this is a ground breaking technique.

**Weaknesses:**

There is one thing that I am a bit confused. If the embeddings of win and lose can be decomposed into two parts, i.e.,  shared semantic component and distinctive semantic component, then it seems that the shared component is somewhat "same" for both win and lose answers. Therefore, even if we use the standard DPO, I think the algorithm will still automatically "ignore" this part, and try to optimize the distinctive part, is it?

I certainly understand that FDT algorithm makes this process explicit, which brings a better optimization process. However, it would be better if the author can explain why "throwing away" the shared component, what is the real benefit? Will the original DPO somehow optimize that component? If so, how does it affect the performance of DPO?

**Questions:**

I do not have additional questions.

---

> ### Author Response · Authors · 2024-11-28
>
> Thank you for your thorough and positive review.
> The key difference between DPO and FDT lies in their gradient update rules:
> Equation 7 (Traditional DPO):
> $$
> \begin{aligned}
>     \Delta W[y]=c(x, y_w, y) h(x, y_w^{1:k-1})^\top+c(x, y_l, y) h(x, y_l^{1:k-1})^\top,
> \end{aligned}
> $$
>
> Equation 8 (FDT):
> $$
> \begin{aligned}
>     \Delta W[y]=\alpha(c(x, y_w, y)-c(x, y_l, y))(h(x, y_w^{1:k-1})-h(x, y_l^{1:k-1}))^\top,
> \end{aligned}
> $$
>
> During fine-tuning, the pre-trained weights already handle shared semantic components effectively. FDT explicitly focuses on distinctive features through its difference-based update rule, while DPO must implicitly learn to distinguish these components. This targeted approach is particularly important for mathematical reasoning, where critical differences often exist within similar contexts. Our theoretical results (Theorem 2) provide formal guarantees that FDT bounds the influence of shared components while amplifying distinctive features, leading to more efficient optimization of discriminative aspects.

---

### Official Review · Reviewer_gz24 · 2024-10-28

**Soundness:** 2
**Presentation:** 2
**Contribution:** 2
**Rating:** 3
**Confidence:** 4

**Summary:**

To enhance the mathematical reasoning abilities of LLMs, this paper proposes the FDT algorithm. The paper points out that traditional DPO algorithms struggle to distinguish correct answers and incorrect answers at the token level, whereas FDT leverages hidden state analysis to fine-tune the model’s output layer weights, achieving better results.

The contributions of this paper include:

1. Proposing the FDT algorithm, which can be plugged into the RLHF framework, to improve mathematical reasoning capabilities.
2. Providing theoretical analysis and formula derivations.
3. Offering experimental results of the FDT algorithm and other baselines.

**Strengths:**

The paper proposes the FDT algorithm to  enhance the mathematical reasoning abilities of LLMs, and provides the detailed formula derivations. Mathematical reasoning capabilities of  LLMs are an important research direction. Considering the difference between correct answers and wrong answers are an interesting perspective. The datasets the paper chooses (e.g. GSM8K and MATH) are widely recognized to evaluate model's math reasoning ability. Experiments are conducted on  Llama-3.2-3B-Instruct and Qwen2.5-3B-Instruct, which both are great open-source LLMs.

**Weaknesses:**

1. Some expressions and statements in this paper are not sufficiently clear, making the article difficult to understand. For example, The expressions in Eq 7 and Eq 8 do not fully correspond to each other.

2. Theorem 2 in this paper, which serves as the foundation of the methodology, does not seem to be entirely correct. There is no evidence to support that the hidden states for correct and incorrect answers should be very close. More often, we prefer the difference between the two vectors to be significantly large. In addition, the similar and different aspects between correct and incorrect answers are difficult to be decoupled. The method in this paper does not provide any effective strategies for decoupling them, but rather assumes that they can be decoupled directly.

3. The experimental results cannot support the effectiveness of the method. Firstly, the experiments do not provide valid ablation analysis to demonstrate the effectiveness of the proposed modules. Secondly, the performance improvements of the proposed method is not significant. Finally, this work does not compare with other baselines. Methods based on LLM have already achieved better performance on the selected benchmarks.

4. The paper does not provide any interpretability results to support its conclusions. The detailed case analysis should be provided to expain how hidden state correction or  difference between the correct answers and wrong answers can influnce mathematical reasoning.

**Questions:**

1. The paper define $y$ as a sequence of tokens in line 197 to 198, but claims token $y$ in line 207 to 209. There are the two inconsistent definitions. Could you clarify what $y$ refers to?

2. There seems to be a lack of continuity between Equation 7 and Equation 8. Could you please provide further explanation?

3. The paper claims " As the responses of the same question are similar, the hidden states of correct response and incorrect response are similar", which seems to be problematic. I believe the authors need to provide more substantial evidence to demonstrate the causal relationship between the two.

4. Since the premise of Theorem 2 is problematic, it raises the question about the validity of Theorem 2 itself. Could you further clarify this issue?

5. The paper adopt the inconsistent  experimental settings between FDT and baselines, such as learning rate and hyperparameter β (in Section 5.3). Why is that?

6. For main results, shown in Table 1, there are several quetions:
(1)  llama-3.2 model fails in  MMLU-redux datasets, only achieving 7.48 % accuracy. Why is that?
(2) The performance of DPO+FDT and Step-DPO+FDT is inconsistent. For example, on the GSM8K dataset, Qwen2.5-3B-Instruct+DPO+FDT improved by only 0.1% over Qwen2.5-3B-Instruct+DPO, while Qwen2.5-3B-Instruct+Step-DPO+FDT showed a 2.4% improvement over Qwen2.5-3B-Instruct+Step-DPO. Could you explain the reason behind this inconsistency?
(3) Following up on the above question, this inconsistency varies across different datasets (GSM8K or MATH) and base models (Qwen2.5 or Llama-3.2). Could you explain the reason for this?
(4) Performance of Llama-3.2-3B-Instruct+DPO+FDT is even lower than Llama-3.2-3B-Instruct+SFT. Why is that?

7. Several minor errors. For example, citation errors in Seciton 5.1; repeated formula in Equation 7; the reference to $K$ is inconsistent in Algorithm 1.

---

> ### Author Response · Authors · 2024-11-28
>
> We thank the reviewer for their detailed feedback. We address each main concern below:
> > The relationship between Eq 7 and Eq 8
>
> We would like to clarify that these two equations serve different purposes and are not meant to be mathematically equivalent. Equation 7 shows the standard DPO gradient update, which represents how DPO traditionally updates the weight matrix by considering both correct and incorrect responses independently. Equation 8 presents our proposed FDT update rule, which is intentionally designed to be different from Equation 7, as it explicitly focuses on amplifying the differences between correct and incorrect responses. Rather than treating them independently (as in Equation 7).
>
> > The assumption in Theorem 2 indicates that the hidden states for correct and incorrect answers should be very close.
>
> We would like to clarify an important misunderstanding about our assumptions and methodology.
> Our assumption is not that "hidden states for correct and incorrect answers should be very close." Rather, we only assume similarity in their norms ($|\mu_+|^2-|\mu_-|^2\leq \delta$), not in their vector representations.
> We have strong empirical evidence supporting this assumption. Our analysis shows that the relative differences in hidden state norms between correct and incorrect responses are consistently small in Appendix B, Figure 3. This figure illustrates the distribution of these relative differences across all sample pairs for two different models: Qwen2.5-3B-Instruct and Mistral-7B-Instruct-v0.3. The histograms reveal that the relative differences are predominantly concentrated around zero for both models. Qwen2.5-3B-Instruct exhibits a mean of 0.0321 and a median of 0.0220, while Mistral-7B-Instruct-v0.3 shows even smaller differences with a mean of 0.0100 and a median of 0.0081. These consistently small differences suggest that the samples maintain similar representation norms in the models' hidden state spaces, regardless of their chosen or rejected labels.
>
> > In addition, the similar and different aspects between correct and incorrect answers are difficult to be decoupled. The method in this paper does not provide any effective strategies for decoupling them, but rather assumes that they can be decoupled directly.
>
> <!-- There have been several recent works that have shown that the hidden states of correct and incorrect answers can be decomposed into two parts [1,2].  -->
>
> Recent research provides strong evidence that hidden states differences can effectively decouple semantic features:
>
> 1. Turner et al. (2024) demonstrate that taking differences between hidden states is an effective way to isolate specific semantic components. They show that their "steering vectors" (computed as activation differences) can selectively modify specific semantic features while preserving others [1]. For example, their sentiment steering vector successfully shifts text sentiment without affecting other content - directly evidencing semantic decoupling. Their comparison with random vectors shows that these differences capture targeted semantic changes rather than broad distributional shifts.
> 2. Marks and Tegmark (2024) provide additional evidence that hidden state differences in language models effectively isolate semantic components [2]. Their experiments reveal that "the model computed and linearly represented some feature which correlates with truth on both cities and neg_cities but with opposite signs" (Section 4.1). This demonstrates that taking differences between hidden states can extract specific semantic features (like truthfulness) while preserving other aspects.
>
> Similar conclusions have been drawn in other works [3,4].
>
> > the experiments do not provide valid ablation analysis to demonstrate the effectiveness of the proposed modules
>
> 1. Regarding ablation studies: Our method is designed as a plug-in enhancement that can be integrated into any pairwise preference optimization framework. The baseline methods (DPO, Step-DPO) effectively serve as ablation studies - they are identical to our approach but without the FDT component. The performance improvements over these baselines directly demonstrate FDT's effectiveness.
>
> 2. Concerning significance of improvements: As another reviewer noted, while not groundbreaking, our method provides "a very useful gadget for doing math reasoning" that can be widely applied. The consistent improvements across multiple models (Llama-3.2-3B-Instruct and Qwen2.5-3B-Instruct) and different base methods (DPO, Step-DPO) demonstrate the robustness of our approach.

---

> > ### Author Response · Authors · 2024-11-28
> >
> > > The paper adopt the inconsistent experimental settings between FDT and baselines, such as learning rate and hyperparameter β (in Section 5.3). Why is that?
> >
> >
> > For a fair comparison, we conducted hyperparameter tuning for each method independently. The different learning rates and $\beta$ values reported in Section 5.3 represent the optimal configurations found for each approach.
> > It would be unfair to use identical hyperparameters across all methods since different algorithms may require different optimal settings to achieve their best performance. This is standard practice in machine learning research - comparing methods at their respective optimal configurations rather than using identical hyperparameters that might favor one approach over another.
> >
> > > llama-3.2 model fails in MMLU-redux datasets, only achieving 7.48 % accuracy. Why is that?
> >
> > We believe this is primarily an artifact of the quantization process used in LLAMA-3.2's training [5] rather than a limitation of our method. The phenomenon appears only on MMLU-redux while our method performs well across other datasets, suggesting this is specific to the base model rather than our approach.
> >
> > > Performance of Llama-3.2-3B-Instruct+DPO+FDT is even lower than Llama-3.2-3B-Instruct+SFT.
> >
> > This observation is consistent with current research trends. Recent work has shown that DPO may hurt performance in some cases [6]. It is believed that the DPO optimization objective may not be effectively increasing the likelihood of preferred sequences despite increasing the reward margin [6, 7]. This can hinder learning from math preference pairs where changing one token can flip the label (e.g., changing 2 + 2 = 4 to 2 + 2 = 5).
> >
> >
> >
> >
> > [1] Marks S, Tegmark M. The geometry of truth: Emergent linear structure in large language model representations of true/false datasets[J]. arXiv preprint arXiv:2310.06824, 2023.
> >
> > [2] Turner A M, Thiergart L, Leech G, et al. Activation addition: Steering language models without optimization[J]. arXiv e-prints, 2023: arXiv: 2308.10248.
> >
> > [3] Liu W, Wang X, Wu M, et al. Aligning large language models with human preferences through representation engineering[J]. arXiv preprint arXiv:2312.15997, 2023.
> >
> > [4] Zou A, Phan L, Chen S, et al. Representation engineering: A top-down approach to ai transparency[J]. arXiv preprint arXiv:2310.01405, 2023.
> >
> > [5] https://huggingface.co/meta-llama/Llama-3.2-1B
> >
> > [6] Meng Y, Xia M, Chen D. Simpo: Simple preference optimization with a reference-free reward[J]. arXiv preprint arXiv:2405.14734, 2024.
> >
> > [7] Pal A, Karkhanis D, Dooley S, et al. Smaug: Fixing failure modes of preference optimisation with dpo-positive[J]. arXiv preprint arXiv:2402.13228, 2024.

---

> > > ### Comment · Reviewer_gz24 · 2024-12-02
> > >
> > > Thanks for your further explanation of your theories and experiments. My concerns are on that your experiments cannot prove the significance of your assumption. As you said "We have strong empirical evidence supporting this assumption", but only empirical evidence cannot support its effectiveness on the tasks. Actually, the results do not provide sufficient evidence of the priority of the methods. On MMLU-redux datasets, I think achieving 7.48 % accuracy means the failure of the task. Therefore, I would like to keep the score.

---

### Official Review · Reviewer_AkPm · 2024-11-04

**Soundness:** 3
**Presentation:** 3
**Contribution:** 2
**Rating:** 5
**Confidence:** 3

**Summary:**

Current alignment methods like DPO often struggle to effectively differentiate between correct and erroneous mathematical  responses. Particularly since there are some shared semantics and some distinctive semantics between responses, they fail to disentangle these two characteristics when imposing the loss at the final token or logit level. Instead this paper proposes to leverage the rich semantic information embedded in the hidden state space of LLMs. Their method Focused Differentiation Training finetunes the model by emphasising the differences between the hidden states of correct and incorrect responses, rather than their common features. The authors provide theoretical analysis as well as some experiments on GSM8k, Math and MMLU-redux datasets.

**Strengths:**

- This an interesting thought proposition to utilise the rich semantic information in the hidden state representations, rather than the final token or logit level analysis.
- Decomposing the hidden states into shared and distinctive semantic component and amplifying  the hidden states that contribute more to the differences in order to improve the models ability to distinguish between correct and incorrect responses seems quite meaningful
- The authors have gone into depths of this problem, giving some theoretical insights and proofs
- Main contribution of the paper are some theoretical analysis which are contingent on some assumptions. I am not sure how general those assumptions are (see Questions to Author)

**Weaknesses:**

- Experiments are not thorough or mature enough. Experimental results are not very convincing as the improvements do not seem very consistent - In most of the cases improvement is around 1-2% which is somewhat insignificant. In some cases improvement is 3-4% or even over 6% for model for either DPO/Step-DPO setup, but if the model or DPO is changed, the performance improvement drops to again ~<1% or even hurting performance in some cases. From all this it is not very clear what is giving performance improvement and under what conditions? Some ablations would definitely be helpful in understand this.
- This seems to be a generic direction to investigate but I am curious why the entire setting in the paper was framed as a problem for mathematical reasoning alone? The authors show some results on MMLU-redux but those are also not very convincing (related to my above point)
- Since this exploration is based on the hidden states rather than the final logics, this seems to be a very model specific characteristic & empirical results also indicate that. Given that more llms should be considered for experiments on this work.
- Overall I feel this can be a good fundamental contribution if more empirical results and better consistency can be shown - more experiments on a wider set of llms across different  sizes and across different tasks - can focus on specific downstream tasks like reasoning (logical, mathematical and planning). At this stage, the paper feels quite incomplete mainly because of lack of enough experiments

**Questions:**

- Does the assumption in Theorem 2 require the hidden states to be bounded with some constant M and \delta for every input? If it is for every input then it seems to be quite restrictive. Remark 1 also then follows the similar restriction Can the authors comment on how general the applicability of this theorem would be?

---

> ### Author Response · Authors · 2024-11-28
>
> We sincerely thank the reviewer for their thorough and constructive feedback. We address each main concern below:
>
> > Experiments are not thorough or mature enough.
> 1. We have conducted additional experiments with more LLM variants including deepseek-llm-7b-chat and Mistral-7B-Instruct-v0.
>
> | Model | GSM8K | MATH |
> |-------|--------|-------|
> | deepseek-llm-7b-chat + DPO | 77.26 | 47.3 |
> | deepseek-llm-7b-chat + DPO + FDT | 77.63 (+0.37) | 49.6 (+2.3) |
> | deepseek-llm-7b-chat + Step-DPO | 76.50 | 46.2 |
> | deepseek-llm-7b-chat + Step-DPO + FDT | 78.47 (+1.97) | 48.5 (+2.3) |
> | Mistral-7B-Instruct-v0 + DPO | 46.86 | 20.88 |
> | Mistral-7B-Instruct-v0 + DPO + FDT | 47.26 (+0.4) | 21.34 (+0.46) |
> | Mistral-7B-Instruct-v0 + Step-DPO | 46.52 | 21.08 |
> | Mistral-7B-Instruct-v0 + Step-DPO + FDT | 46.80 (+0.28) | 21.26 (+0.18) |
>
> 2. Our method is designed as a plug-in enhancement that can be integrated into any pairwise preference optimization framework. The baseline methods (DPO, Step-DPO) effectively serve as ablation studies - they are identical to our approach but without the FDT component. The performance improvements over these baselines directly demonstrate FDT's effectiveness.
> 3. As another reviewer noted, while not groundbreaking, our method provides "a very useful gadget for doing math reasoning" that can be widely applied. The consistent improvements across multiple models (Llama-3.2-3B-Instruct and Qwen2.5-3B-Instruct) and different base methods (DPO, Step-DPO) demonstrate the robustness of our approach.
>
> > The reason why the entire setting in the paper was framed as a problem for mathematical reasoning alone
>
> Mathematical reasoning presents an ideal test case for our method because it involves subtle semantic differences that can dramatically change the correctness of a response. In mathematical problem-solving:
>
> 1. Small semantic variations often lead to critically different outcomes. For example, the difference between "greater than" versus "greater than or equal to" can completely change a solution's validity. Our method specifically addresses these fine-grained semantic distinctions through focused differentiation of hidden states.
> 2. Mathematical reasoning provides clear ground truth for evaluating semantic differences. Unlike more subjective tasks, mathematical problems have definitive correct and incorrect answers, making them perfect for validating our approach to semantic differentiation.
> 3. The structured nature of mathematical language allows us to isolate and analyze how our method handles semantic differences in a controlled setting. This is particularly valuable for demonstrating the effectiveness of our hidden state differentiation approach.
>
> > Does the assumption in Theorem 2 require the hidden states to be bounded with some constant $M$ and $\delta$ for every input?
>
> No, the assumption does not require the hidden states to be bounded with some constant $M$ and $\delta$ for every input. The assumption requires that the mean of the hidden states of correct and incorrect responses is bounded. We have provided empirical evidence supporting this assumption in Appendix B, Figure 3, which demonstrates that this assumption is generally valid across different models

---

> > ### Comment · Reviewer_AkPm · 2024-11-29
> >
> > Thank you for the further elaboration and experiments. But the performance improvement is not significant enough. Based on this i still think it requires more work to establish the fundamental contribtion of this method and would like to keep my original score.

---

### Official Review · Reviewer_32GV · 2024-11-04

**Soundness:** 2
**Presentation:** 2
**Contribution:** 2
**Rating:** 3
**Confidence:** 2

**Summary:**

This paper proposes a tweak to DPO-style post-training for large language models. In particular, their technique can be applied when a dataset of paired examples $(x, y_w, y_l)$ is available (where $y_w$ is a preferred completion and $y_l$ is a dispreferred completion). The proposal is to modify the gradient update for the last-layer weight matrix. The usual gradient update for the last-layer weight matrix is a function of the last-layer hidden representations of $y_w$ and $y_l$. The proposed modification makes the update a function only of the \textit{difference} in these representations. The paper shows how their proposed method can be implemented using stop-gradient/detach, and illustrates that it improves performance on mathematical reasoning problems.

**Strengths:**

The main strength of the paper is that the proposed method appears to have some positive effect, empirically. Fig. 2 suggests that the method does help the model better distinguish between preferred and dispreferred responses, and the results in Tab. 1 suggest that the method might be useful for marginally improving performance on some reasoning tasks.

**Weaknesses:**

The proposed approach is described and motivated rather vaguely. The goal is to "focus on" semantic differences. But there isn't really any explanation or analysis of why the proposed approach facilitates this, or in what sense it "focuses" on the differences. The method itself is explained in terms of a series of stop-gradient/detach operations; it is not clear what effect this series of operations has on the objective function being optimized or the dynamics of training. No intuition is provided for why semantic differences should be computed only at the last layer. The theorems are not unpacked to explain why they are interesting or why they validate core claims of the paper. As for the empirical results, they are encouraging, but there is no real empirical analysis of why the method helps, alternative design choices, etc. -- just benchmark numbers that go up.

**Questions:**

The related work section talks only generally about DPO and mathematical reasoning. Are there any techniques that are more closely related to your proposed approach?

Minor comment: the equations on L201 and L203 are missing the probability of y_1.

---

> ### Author Response · Authors · 2024-11-28
>
> We appreciate the reviewer's thoughtful comments about the need for clearer motivation and analysis of our approach. We address these concerns by providing additional explanations and evidence:
>
> 1. Theoretical Foundation for Hidden State Differences
>
>     Our approach of focusing on hidden state differences is well-grounded in recent research on semantic feature isolation. Turner et al. (2024) demonstrated that differences between hidden states effectively isolate specific semantic components while preserving others. Their work shows that computing differences between hidden states can selectively modify specific semantic features without affecting other content - providing direct evidence for semantic decoupling. This aligns with our method's goal of amplifying the distinctive semantic components that differentiate correct from incorrect mathematical reasoning.
>
> 2. Mechanism of Semantic Differentiation
>
>     The key distinction between traditional DPO (Equation 7) and our FDT approach (Equation 8) lies in how they handle semantic information:
>
>     Equation 7 (Traditional DPO):
>     $$
>     \begin{aligned}
>         \Delta W[y]=c(x, y_w, y) h(x, y_w^{1:k-1})^\top+c(x, y_l, y) h(x, y_l^{1:k-1})^\top,
>     \end{aligned}
>     $$
>
>     Equation 8 (FDT):
>     $$
>     \begin{aligned}
>         \Delta W[y]&=\alpha(c(x, y_w, y)-c(x, y_l, y))(h(x, y_w^{1:k-1})-h(x, y_l^{1:k-1}))^\top,
>     \end{aligned}
>     $$
>
>     While DPO treats correct and incorrect responses independently, FDT explicitly computes their difference to focus on discriminative features. This is not merely a mathematical transformation - it fundamentally changes what the model learns to attend to.
>
> 3. Last Layer Focus and Theoretical Guarantees
>
>     We focus on the last layer's hidden states because they directly influence the model's logits through the output layer weight matrix W. This direct connection means that modifying the last layer's representations has the most immediate impact on the model's predictions. Our theoretical results formalize how this modification works:
>
>   - Theorem 2 guarantees that our update rule bounds the influence of shared semantic components: $\Delta W[y](h(y|x, y_w^{1:k-1})+h(y|x, y_l^{1:k-1}))\le 4\alpha\delta$. This means that features common to both correct and incorrect responses have limited impact on the model's predictions.
>
>   - Corollary 1 ensures that our method effectively amplifies the distinctive features: $\Delta W[y](h(y|x, y_w^{1:k-1})-h(y|x, y_l^{1:k-1}))\ge \frac{1}{2}\alpha\|\mu_+-\mu_-\|$. This theoretical guarantee shows that the features that distinguish correct from incorrect responses are emphasized in the model's decision-making process.
>
>     Together, these results demonstrate that by focusing on the last layer, where hidden states directly map to predictions through W, we can effectively control what information influences the model's outputs - suppressing shared features while amplifying discriminative ones. While we directly modify the last layer's hidden states, the gradient backpropagation through the model ensures that earlier layers are also optimized to better capture and propagate discriminative features.

---

> > ### Comment · Reviewer_32GV · 2024-12-03
> >
> > Thanks for your response. At a glance I am still a bit confused because your English glosses of the theorems focus on "the model's predictions" or "the model's decision-making process," but the theorems themselves are about an update rule, not the (final, trained) model's behavior.
> >
> > I still feel, as in the original review, that it is not clear what effect your stop-gradient/detach operations have on the objective being optimized or the dynamics of training. As such, I maintain my current score.

---

### Meta-Review · Area_Chair_acF7 · 2024-12-17

**Metareview:**

This paper focuses on fine-tuning LLMs for mathematical reasoning and proposes changing the DPO training procedure by using information from the intermediate activations of a Transformer in order to focus on “semantic differences”. The experimental results suggest improvements over baselines. As the reviewers also point out, the experiments show marginal improvements and don’t convincingly show why the proposed method helps – if all we know is that the final metric goes up a little bit, we can’t draw strong conclusions about the proposed methods. Ablations showing why the method helps would be helpful.

**Additional Comments On Reviewer Discussion:**

See above. Additionally, the reviewers found parts of the paper unclear and possibly incorrect. The authors provided additional experiments but they were not convincing enough for the reviewers.

---

### Decision · Program_Chairs · 2025-01-22

Reject